# Exploiting Linear Structure Within Convolutional Networks for Efficient Evaluation

**Emily Denton, Wojciech Zaremba, Joan Bruna, Yann LeCun and Rob Fergus**

Dept. of Computer Science, Courant Institute, New York University
{denton, zaremba, bruna, lecun, fergus} @cs.nyu.edu

## Abstract

We present techniques for speeding up the test-time evaluation of large convolutional networks, designed for object recognition tasks. These models deliver impressive accuracy, but each image evaluation requires millions of floating point operations, making their deployment on smartphones and Internet-scale clusters problematic. The computation is dominated by the convolution operations in the lower layers of the model. We exploit the redundancy present within the convolutional filters to derive approximations that significantly reduce the required computation. Using large state-of-the-art models, we demonstrate speedups of convolutional layers on both CPU and GPU by a factor of $2\times$, while keeping the accuracy within $1\%$ of the original model.

## 1 Introduction

Large neural networks have recently demonstrated impressive performance on a range of speech and vision tasks. However, the size of these models can make their deployment at test time problematic. For example, mobile computing platforms are limited in their CPU speed, memory and battery life. At the other end of the spectrum, Internet-scale deployment of these models requires thousands of servers to process the 100's of millions of images per day. The electrical and cooling costs of these servers is significant. Training large neural networks can take weeks, or even months. This hinders research and consequently there have been extensive efforts devoted to speeding up training procedure. However, there are relatively few efforts aimed at improving the *test-time* performance of the models.

We consider convolutional neural networks (CNNs) used for computer vision tasks, since they are large and widely used in commercial applications. These networks typically require a huge number of parameters ($\sim 10^8$ in [1]) to produce state-of-the-art results. While these networks tend to be hugely over parameterized [2], this redundancy seems necessary in order to overcome a highly non-convex optimization [3]. As a byproduct, the resulting network wastes computing resources. In this paper we show that this redundancy can be exploited with linear compression techniques, resulting in significant speedups for the evaluation of *trained* large scale networks, with minimal compromise to performance.

We follow a relatively simple strategy: we start by compressing each convolutional layer by finding an appropriate low-rank approximation, and then we fine-tune the upper layers until the prediction performance is restored. We consider several elementary tensor decompositions based on singular value decompositions, as well as filter clustering methods to take advantage of similarities between learned features.

Our main contributions are the following: (1) We present a collection of generic methods to exploit the redundancy inherent in deep CNNs. (2) We report experiments on state-of-the-art Imagenet

CNNs, showing empirical speedups on convolutional layers by a factor of $2 - 3\times$ and a reduction of parameters in fully connected layers by a factor of $5 - 10\times$.

**Notation:** Convolution weights can be described as a 4-dimensional tensor: $W \in \mathbb{R}^{C \times X \times Y \times F}$. $C$ is the number of number of input channels, $X$ and $Y$ are the spatial dimensions of the kernel, and $F$ is the target number of feature maps. It is common for the first convolutional layer to have a stride associated with the kernel which we denote by $\Delta$. Let $I \in \mathbb{R}^{C \times N \times M}$ denote an input signal where $C$ is the number of input maps, and $N$ and $M$ are the spatial dimensions of the maps. The target value, $T = I * W$, of a generic convolutional layer, with $\Delta = 1$, for a particular output feature, $f$, and spatial location, $(x, y)$, is

$$T(f, x, y) = \sum_{c=1}^{C} \sum_{x'=1}^{X} \sum_{y'=1}^{Y} I(c, x - x', y - y') W(c, x', y', f)$$

If $W$ is a tensor, $\|W\|$ denotes its operator norm, $\sup_{\|x\|=1} \|Wx\|_F$ and $\|W\|_F$ denotes its Frobenius norm.

## 2   Related Work

Vanhoucke *et al.* [4] explored the properties of CPUs to speed up execution. They present many solutions specific to Intel and AMD CPUs and some of their techniques are general enough to be used for any type of processor. They describe how to align memory, and use SIMD operations (vectorized operations on CPU) to boost the efficiency of matrix multiplication. Additionally, they propose the linear quantization of the network weights and input. This involves representing weights as 8-bit integers (range $[-127, 128]$), rather than 32-bit floats. This approximation is similar in spirit to our approach, but differs in that it is applied to each weight element independently. By contrast, our approximation approach models the structure within each filter. Potentially, the two approaches could be used in conjunction.

The most expensive operations in CNNs are the convolutions in the first few layers. The complexity of this operation is linear in the area of the receptive field of the filters, which is relatively large for these layers. However, Mathieu *et al.* [5] have shown that convolution can be efficiently computed in Fourier domain, where it becomes element-wise multiplication (and there is no cost associated with size of receptive field). They report a forward-pass speed up of around $2\times$ for convolution layers in state-of-the-art models. Importantly, the FFT method can be used jointly with most of the techniques presented in this paper.

The use of low-rank approximations in our approach is inspired by work of Denil *et al.* [2] who demonstrate the redundancies in neural network parameters. They show that the weights within a layer can be accurately predicted from a small (e.g. $\sim 5\%$) subset of them. This indicates that neural networks are heavily over-parametrized. All the methods presented here focus on exploiting the linear structure of this over-parametrization.

Finally, a recent preprint [6] also exploits low-rank decompositions of convolutional tensors to speed up the evaluation of CNNs, applied to scene text character recognition. This work was developed simultaneously with ours, and provides further evidence that such techniques can be applied to a variety of architectures and tasks. Our work differs in several ways. First, we consider a significantly larger model. This makes it more challenging to compute efficient approximations since there are more layers to propagate through and thus a greater opportunity for error to accumulate. Second, we present different compression techniques for the hidden convolutional layers and provide a method of compressing the first convolutional layer. Finally, we present GPU results in addition to CPU results.

## 3   Convolutional Tensor Compression

In this section we describe techniques for compressing 4 dimensional convolutional weight tensors and fully connected weight matrices into a representation that permits efficient computation and storage. Section 3.1 describes how to construct a good approximation criteria. Section 3.2 describes

techniques for low-rank tensor approximations. Sections 3.3 and 3.4 describe how to apply these techniques to approximate weights of a convolutional neural network.

## 3.1 Approximation Metric

Our goal is to find an approximation, $\tilde{W}$, of a convolutional tensor $W$ that facilitates more efficient computation while maintaining the prediction performance of the network. A natural choice for an approximation criterion is to minimize $\|\tilde{W} - W\|_F$. This criterion yields efficient compression schemes using elementary linear algebra, and also controls the operator norm of each linear convolutional layer. However, this criterion assumes that all directions in the space of weights equally affect prediction performance. We now present two methods of improving this criterion while keeping the same efficient approximation algorithms.

**Mahalanobis distance metric:** The first distance metric we propose seeks to emphasize coordinates more prone to produce prediction errors over coordinates whose effect is less harmful for the overall system. We can obtain such measurements as follows. Let $\Theta = \{W_1, \ldots, W_S\}$ denote the set of all parameters of the $S$-layer network, and let $U(I; \Theta)$ denote the output after the softmax layer of input image $I$. We consider a given input training set $(I_1, \ldots, I_N)$ with known labels $(y_1, \ldots, y_N)$. For each pair $(I_n, y_n)$, we compute the forward propagation pass $U(I_n, \Theta)$, and define as $\{\beta_n\}$ the indices of the $h$ largest values of $U(I_n, \Theta)$ different from $y_n$. Then, for a given layer $s$, we compute

$$d_{n,l,s} = \nabla_{W_s} \left( U(I_n, \Theta) - \delta(i - l) \right) , \ n \leq N , \ l \in \{\beta_n\} , \ s \leq S , \tag{1}$$

where $\delta(i-l)$ is the dirac distribution centered at $l$. In other words, for each input we back-propagate the difference between the current prediction and the $h$ "most dangerous" mistakes.

The Mahalanobis distance is defined from the covariance of $d$: $\|W\|_{maha}^2 = w\Sigma^{-1}w^T$, where $w$ is the vector containing all the coordinates of $W$, and $\Sigma$ is the covariance of $(d_{n,l,s})_{n,l}$. We do not report results using this metric, since it requires inverting a matrix of size equal to the number of parameters, which can be prohibitively expensive in large networks. Instead we use an approximation that considers only the diagonal of the covariance matrix. In particular, we propose the following, approximate, Mahalanobis distance metric:

$$\|W\|_{\widetilde{maha}} := \sum_p \alpha_p W(p) , \text{ where } \alpha_p = \left( \sum_{n,l} d_{n,l,s}(p)^2 \right)^{1/2} \tag{2}$$

where the sum runs over the tensor coordinates. Since (2) is a reweighted Euclidean metric, we can simply compute $W' = \alpha \ .* \ W$, where $.*$ denotes element-wise multiplication, then compute the approximation $\tilde{W}'$ on $W'$ using the standard $L_2$ norm, and finally output $\tilde{W} = \alpha^{-1} .* \tilde{W}'$.

**Data covariance distance metric:** One can view the Frobenius norm of $W$ as $\|W\|_F^2 = \mathbb{E}_{x \sim \mathcal{N}(0,I)} \|Wx\|_F^2$. Another alternative, similar to the one considered in [6], is to replace the isotropic covariance assumption by the empirical covariance of the input of the layer. If $W \in \mathbb{R}^{C \times X \times Y \times F}$ is a convolutional layer, and $\widehat{\Sigma} \in \mathbb{R}^{CXY \times CXY}$ is the empirical estimate of the input data covariance, it can be efficiently computed as

$$\|W\|_{data} = \|\widehat{\Sigma}^{1/2} W_F\|_F , \tag{3}$$

where $W_F$ is the matrix obtained by folding the first three dimensions of $W$. As opposed to [6], this approach adapts to the input distribution without the need to iterate through the data.

## 3.2 Low-rank Tensor Approximations

### 3.2.1 Matrix Decomposition

Matrices are 2-tensors which can be linearly compressed using the Singular Value Decomposition. If $W \in \mathbb{R}^{m \times k}$ is a real matrix, the SVD is defined as $W = USV^\top$, where $U \in \mathbb{R}^{m \times m}, S \in \mathbb{R}^{m \times k}, V \in \mathbb{R}^{k \times k}$. $S$ is a diagonal matrix with the singular values on the diagonal, and $U$, $V$ are orthogonal matrices. If the singular values of $W$ decay rapidly, $W$ can be well approximated by keeping only the $t$ largest entries of $S$, resulting in the approximation $\tilde{W} = \tilde{U}\tilde{S}\tilde{V}^\top$, where

$\tilde{U} \in \mathbb{R}^{m \times t}, \tilde{S} \in \mathbb{R}^{t \times t}, \tilde{V} \in \mathbb{R}^{t \times k}$ Then, for $I \in \mathbb{R}^{n \times m}$, the approximation error $\|I\tilde{W} - IW\|_F$ satisfies $\|I\tilde{W} - IW\|_F \leq s_{t+1}\|I\|_F$ , and thus is controlled by the decay along the diagonal of $S$. Now the computation $I\tilde{W}$ can be done in $O(nmt + nt^2 + ntk)$, which, for sufficiently small $t$ is significantly smaller than $O(nmk)$.

### 3.2.2 Higher Order Tensor Approximations

SVD can be used to approximate a tensor $W \in \mathbb{R}^{m \times n \times k}$ by first folding all but two dimensions together to convert it into a 2-tensor, and then considering the SVD of the resulting matrix. For example, we can approximate $W_m \in \mathbb{R}^{m \times (nk)}$ as $\tilde{W}_m \approx \tilde{U}\tilde{S}\tilde{V}^\top$. $W$ can be compressed even further by applying SVD to $\tilde{V}$. We refer to this approximation as the SVD decomposition and use $K_1$ and $K_2$ to denote the rank used in the first and second application of SVD respectively.

Alternatively, we can approximate a 3-tensor, $W_S \in \mathbb{R}^{m \times n \times k}$, by a rank 1 3-tensor by finding a decomposition that minimizes

$$\|W - \alpha \otimes \beta \otimes \gamma\|_F , \tag{4}$$

where $\alpha \in \mathbb{R}^m$, $\beta \in \mathbb{R}^n$, $\gamma \in \mathbb{R}^k$ and $\otimes$ denotes the outer product operation. Problem (4) is solved efficiently by performing alternate least squares on $\alpha$, $\beta$ and $\gamma$ respectively, although more efficient algorithms can also be considered [7].

This easily extends to a rank $K$ approximation using a greedy algorithm: Given a tensor $W$, we compute $(\alpha, \beta, \gamma)$ using (4), and we update $W^{(k+1)} \leftarrow W^k - \alpha \otimes \beta \otimes \gamma$. Repeating this operation $K$ times results in

$$\tilde{W}_S = \sum_{k=1}^{K} \alpha_k \otimes \beta_k \otimes \gamma_k . \tag{5}$$

We refer to this approximation as the outer product decomposition and use $K$ to denote the rank of the approximation.

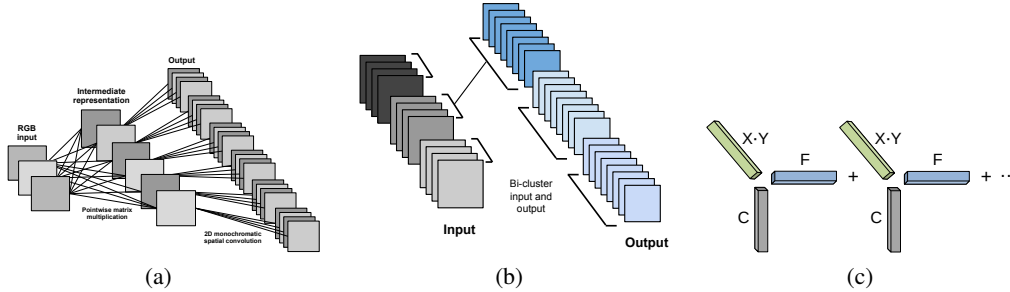

| (a) | (b) | (c) |

Figure 1: A visualization of monochromatic and biclustering approximation structures. **(a)** The monochromatic approximation, used for the first layer. Input color channels are projected onto a set of intermediate color channels. After this transformation, output features need only to look at one intermediate color channel. **(b)** The biclustering approximation, used for higher convolution layers. Input and output features are clustered into equal sized groups. The weight tensor corresponding to each pair of input and output clusters is then approximated. **(c)** The weight tensors for each input-output pair in (b) are approximated by a sum of rank 1 tensors using techniques described in 3.2.2

## 3.3 Monochromatic Convolution Approximation

Let $W \in \mathbb{R}^{C \times X \times Y \times F}$ denote the weights of the first convolutional layer of a trained network. We found that the color components of trained CNNs tend to have low dimensional structure. In particular, the weights can be well approximated by projecting the color dimension down to a 1D subspace. The low-dimensional structure of the weights is illustrated in Figure 4.1.

The monochromatic approximation exploits this structure and is computed as follows. First, for every output feature, $f$, we consider the matrix $W_f \in \mathbb{R}^{C \times (XY)}$, where the spatial dimensions of the filter corresponding to the output feature have been combined, and find the SVD, $W_f = U_f S_f V_f^\top$,

| Approximation technique | Number of operations |
|---|---|
| No approximation | $XYCFNM\Delta^{-2}$ |
| Monochromatic | $C'CNM + XYFNM\Delta^{-2}$ |
| Biclustering + outer product decomposition | $GHK(NM\frac{C}{G} + XYNM\Delta^{-2} + \frac{F}{H}NM\Delta^{-2})$ |
| Biclustering + SVD | $GHNM(\frac{C}{G}K_1 + K_1XYK_2\Delta^{-2} + K_2\frac{F}{H})$ |

Table 1: Number of operations required for various approximation methods.

where $U_f \in \mathbb{R}^{C \times C}, S_f \in \mathbb{R}^{C \times XY}$, and $V_f \in \mathbb{R}^{XY \times XY}$. We then take the rank 1 approximation of $W_f$, $\tilde{W}_f = \tilde{U}_f \tilde{S}_f \tilde{V}_f^\top$ , where $\tilde{U}_f \in \mathbb{R}^{C \times 1}, \tilde{S}_f \in \mathbb{R}, \tilde{V}_f \in \mathbb{R}^{1 \times XY}$. We can further exploit the regularity in the weights by sharing the color component basis between different output features. We do this by clustering the $F$ left singular vectors, $\tilde{U}_f$, of each output feature $f$ into $C'$ clusters, for $C' < F$ . We constrain the clusters to be of equal size as discussed in section 3.4. Then, for each of the $\frac{F}{C'}$ output features, $f$, that is assigned to cluster $c_f$, we can approximate $W_f$ with $\tilde{W}_f = U_{c_f} \tilde{S}_f \tilde{V}_f^\top$ where $U_{c_f} \in \mathbb{R}^{C \times 1}$ is the cluster center for cluster $c_f$ and $\tilde{S}_f$ and $\tilde{V}_f$ are as before.

This monochromatic approximation is illustrated in the left panel of Figure 1(c). Table 1 shows the number of operations required for the standard and monochromatic versions.

### 3.4 Biclustering Approximations

We exploit the redundancy within the 4-D weight tensors in the higher convolutional layers by clustering the filters, such that each cluster can be accurately approximated by a low-rank factorization. We start by clustering the rows of $W_C \in \mathbb{R}^{C \times (XYF)}$, which results in clusters $C_1, \ldots, C_a$. Then we cluster the columns of $W_F \in \mathbb{R}^{(CXY) \times F}$, producing clusters $F_1, \ldots, F_b$. These two operations break the original weight tensor $W$ into $ab$ sub-tensors $\{W_{C_i, F_j}\}_{i=1,\ldots,a, j=1,\ldots,b}$ as shown in Figure 1(b). Each sub-tensor contains similar elements, and thus is easier to fit with a low-rank approximation.

In order to exploit the parallelism inherent in CPU and GPU architectures it is useful to constrain clusters to be of equal sizes. We therefore perform the biclustering operations (or clustering for monochromatic filters in Section 3.3) using a modified version of the $k$-means algorithm which balances the cluster count at each iteration. It is implemented with the Floyd algorithm, by modifying the Euclidean distance with a subspace projection distance.

After the input and output clusters have been obtained, we find a low-rank approximation of each sub-tensor using either the SVD decomposition or the outer product decomposition as described in Section 3.2.2. We concatenate the $X$ and $Y$ spatial dimensions of the sub-tensors so that the decomposition is applied to the 3-tensor, $W_S \in \mathbb{R}^{C \times (XY) \times F}$. While we could look for a separable approximation along the spatial dimensions as well, we found the resulting gain to be minimal. Using these approximations, the target output can be computed with significantly fewer operations. The number of operations required is a function the number of input clusters, $G$, the output clusters $H$ and the rank of the sub-tensor approximations ($K_1, K_2$ for the SVD decomposition; $K$ for the outer product decomposition. The number of operations required for each approximation is described in Table 1.

### 3.5 Fine-tuning

Many of the approximation techniques presented here can efficiently compress the weights of a CNN with negligible degradation of classification performance provided the approximation is not too harsh. Alternatively, one can use a harsher approximation that gives greater speedup gains but hurts the performance of the network. In this case, the approximated layer and all those below it can be fixed and the upper layers can be fine-tuned until the original performance is restored.

## 4 Experiments

We use the 15 layer convolutional architecture of [8], trained on the ImageNet 2012 dataset [9]. The network contains 5 convolutional layers, 3 fully connected layers and a softmax output layer. We

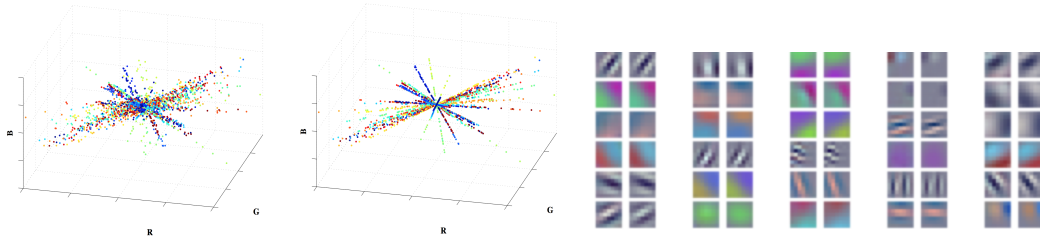

Figure 2: Visualization of the 1st layer filters. **(Left)** Each component of the 96 7x7 filters is plotted in RGB space. Points are colored based on the output filter they belong to. Hence, there are 96 colors and $7^2$ points of each color. Leftmost plot shows the original filters and the right plot shows the filters after the monochromatic approximation, where each filter has been projected down to a line in colorspace. **(Right)** Original and approximate versions of a selection of 1st layer filters.

evaluated the network on both CPU and GPU platforms. All measurements of prediction performance are with respect to the 50K validation images from the ImageNet12 dataset.

We present results showing the performance of the approximations described in Section 3 in terms of prediction accuracy, speedup gains and reduction in memory overhead. All of our fine-tuning results were achieved by training with less than 2 passes using the ImageNet12 training dataset. Unless stated otherwise, classification numbers refer to those of fine-tuned models.

## 4.1 Speedup

The majority of forward propagation time is spent on the first two convolutional layers (see Supplementary Material for breakdown of time across all layers). Because of this, we restrict our attention to the first and second convolutional layers in our speedup experiments. However, our approximations could easily applied to convolutions in upper layers as well.

We implemented several CPU and GPU approximation routines in an effort to achieve empirical speedups. Both the baseline and approximation CPU code is implemented in C++ using Eigen3 library [10] compiled with Intel MKL. We also use Intel's implementation of openmp and multithreading. The baseline gives comparable performance to highly optimized MATLAB convolution routines and all of our CPU speedup results are computed relative to this. We used Alex Krizhevsky's CUDA convolution routines [1] as a baseline for GPU comparisons. The approximation versions are written in CUDA. All GPU code was run on a standard nVidia Titan card.

We have found that in practice it is often difficult to achieve speedups close to the theoretical gains based on the number of arithmetic operations (see Supplementary Material for discussion of theoretical gains). Moreover, different computer architectures and CNN architectures afford different optimization strategies making most implementations highly specific. However, regardless of implementation details, all of the approximations we present reduce both the number of operations and number of weights required to compute the output by at least a factor of two, often more.

### 4.1.1 First Layer

The first convolutional layer has 3 input channels, 96 output channels and 7x7 filters. We approximated the weights in this layer using the monochromatic approximation described in Section 3.3. The monochromatic approximation works well if the color components span a small number of one dimensional subspaces. Figure 2 illustrates the effect of the monochromatic approximation on the first layer filters.

The only parameter in the approximation is $C'$, the number of color channels used for the intermediate representation. As expected, the network performance begins to degrade as $C'$ decreases. The number of floating point operations required to compute the output of the monochromatic convolution is reduced by a factor of $2 - 3\times$, with the larger gain resulting for small $C'$. Figure 3 shows the empirical speedups we achieved on CPU and GPU and the corresponding network performance for various numbers of colors used in the monochromatic approximation. Our CPU and GPU imple-

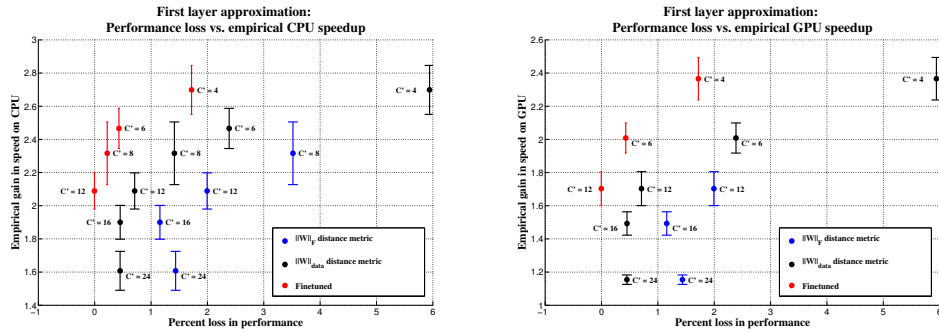

Figure 3: Empirical speedups on (**Left**) CPU and (**Right**) GPU for the first layer. $C'$ is the number of colors used in the approximation.

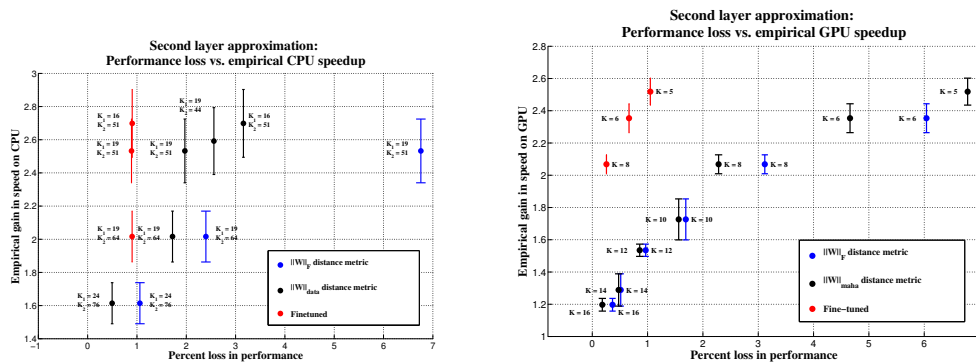

Figure 4: Empirical speedups for second convolutional layer. (**Left**) Speedups on CPU using biclustered ($G = 2$ and $H = 2$) with SVD approximation. (**Right**) peedups on GPU using biclustered ($G = 48$ and $H = 2$) with outer product decomposition approximation.

mentations achieve empirical speedups of $2 - 2.5\times$ relative to the baseline with less than 1% drop in classification performance.

### 4.1.2 Second Layer

The second convolutional layer has 96 input channels, 256 output channels and 5x5 filters. We approximated the weights using the techniques described in Section 3.4. We explored various configurations of the approximations by varying the number of input clusters $G$, the number of output clusters $H$ and the rank of the approximation (denoted by $K_1$ and $K_2$ for the SVD decomposition and $K$ for the outer product decomposition).

Figure 4 shows our empirical speedups on CPU and GPU and the corresponding network performance for various approximation configurations. For the CPU implementation we used the biclustering with SVD approximation. For the GPU implementation we using the biclustering with outer product decomposition approximation. We achieved promising results and present speedups of $2 - 2.5\times$ relative to the baseline with less than a 1% drop in performance.

### 4.2 Combining approximations

The approximations can also be cascaded to provide greater speedups. The procedure is as follows. Compress the first convolutional layer weights and then fine-tune all the layers above until performance is restored. Next, compress the second convolutional layer weights that result from the fine-tuning. Fine-tune all the layers above until performance is restored and then continue the process.

We applied this procedure to the first two convolutional layers. Using the monochromatic approximation with 6 colors for the first layer and the biclustering with outer product decomposition approx-

| Approximation method | Number of parameters | Approximation hyperparameters | Reduction in weights | Increase in error |
|---|---|---|---|---|
| Standard colvolution | $CXYF$ | | | |
| Conv layer 1: Monochromatic | $CC' + XYF$ | $C' = 6$ | 3× | 0.43% |
| Conv layer 2: Biclustering + outer product decomposition | $GHK(\frac{C}{G} + XY + \frac{F}{H})$ | $G = 48; H = 2; K = 6$ | 5.3× | 0.68% |
| Conv layer 2: Biclustering + SVD | $GH(\frac{C}{G}K_1 + K_1 XY K_2 + K_2\frac{F}{H})$ | $G = 2; H = 2; K_1 = 19; K_2 = 24$ | 3.9× | 0.9% |
| Standard FC | $NM$ | | | |
| FC layer 1: Matrix SVD | $NK + KM$ | $K = 250$ <br> $K = 950$ | 13.4× <br> 3.5× | 0.8394% <br> 0.09% |
| FC layer 2: Matrix SVD | $NK + KM$ | $K = 350$ <br> $K = 650$ | 5.8× <br> 3.14× | 0.19% <br> 0.06% |
| FC layer 3: Matrix SVD | $NK + KM$ | $K = 250$ <br> $K = 850$ | 8.1× <br> 2.4× | 0.67% <br> 0.02% |

Table 2: Number of parameters expressed as a function of hyperparameters for various approximation methods and empirical reduction in parameters with corresponding network performance.

imation for the second layer ($G = 48; H = 2; K = 8$) and fine-tuning with a single pass through the training set we are able to keep accuracy within 1% of the original model. This procedure could be applied to each convolutional layer, in this sequential manner, to achieve overall speedups much greater than any individual layer can provide. A more comprehensive summary of these results can be found in the Supplementary Material.

### 4.3 Reduction in memory overhead

In many commercial applications memory conservation and storage are a central concern. This mainly applies to embedded systems (e.g. smartphones), where available memory is limited, and users are reluctant to download large files. In these cases, being able to compress the neural network is crucial for the viability of the product.

In addition to requiring fewer operations, our approximations require significantly fewer parameters when compared to the original model. Since the majority of parameters come from the fully connected layers, we include these layers in our analysis of memory overhead. We compress the fully connected layers using standard SVD as described in 3.2.2, using $K$ to denote the rank of the approximation.

Table 2 shows the number of parameters for various approximation methods as a function of hyperparameters for the approximation techniques. The table also shows the empirical reduction of parameters and the corresponding network performance for specific instantiations of the approximation parameters.

## 5 Discussion

In this paper we have presented techniques that can speed up the bottleneck convolution operations in the first layers of a CNN by a factor $2 - 3\times$, with negligible loss of performance. We also show that our methods reduce the memory footprint of weights in the first two layers by factor of $2 - 3\times$ and the fully connected layers by a factor of $5 - 13\times$. Since the vast majority of weights reside in the fully connected layers, compressing only these layers translates into a significant savings, which would facilitate mobile deployment of convolutional networks. These techniques are orthogonal to other approaches for efficient evaluation, such as quantization or working in the Fourier domain. Hence, they can potentially be used together to obtain further gains.

An interesting avenue of research to explore in further work is the ability of these techniques to aid in regularization either during or post training. The low-rank projections effectively decrease number of learnable parameters, suggesting that they might improve generalization ability. The regularization potential of the low-rank approximations is further motivated by two observations. The first is that the approximated filters for the first conolutional layer appear to be cleaned up versions of the original filters. Additionally, we noticed that we sporadically achieve better test error with some of the more conservative approximations.

### Acknowledgments

The authors are grateful for support from ONR #N00014-13-1-0646, NSF #1116923, #1149633 and Microsoft Research.

## Footnotes

[1] https://code.google.com/p/cuda-convnet/

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
