[Supplementary Material]

# Supplement to "Exploiting Linear Structure Within Convolutional Networks for Efficient Evaluation" NIPS2014

## 1 Forward propagation time breakdown

Table 1 shows the time breakdown of forward propagation for each layer in the CNN architecture we explored. Close to 90% of the time is spent on convolutional layers, and within these layers the majority of time is spent on the first two.

| Layer | Time per batch (sec) | Fraction | Layer | Time per batch (sec) | Fraction |
|---|---|---|---|---|---|
| Conv1 | $2.8317 \pm 0.1030$ | 21.97% | Conv1 | $0.0604 \pm 0.0112$ | 5.14% |
| MaxPool | $0.1059 \pm 0.0154$ | 0.82% | MaxPool | $0.0072 \pm 0.0040$ | 0.61% |
| LRNormal | $0.1918 \pm 0.0162$ | 1.49% | LRNormal | $0.0041 \pm 0.0043$ | 0.35% |
| Conv2 | $4.2626 \pm 0.0740$ | 33.07% | Conv2 | $0.4663 \pm 0.0072$ | 39.68% |
| MaxPool | $0.0705 \pm 0.0029$ | 0.55% | MaxPool | $0.0032 \pm 0.0000$ | 0.27% |
| LRNormal | $0.0772 \pm 0.0027$ | 0.60% | LRNormal | $0.0015 \pm 0.0003$ | 0.13% |
| Conv3 | $1.8689 \pm 0.0577$ | 14.50% | Conv3 | $0.2219 \pm 0.0014$ | 18.88% |
| MaxPool | $0.0532 \pm 0.0018$ | 0.41% | MaxPool | $0.0016 \pm 0.0000$ | 0.14% |
| Conv4 | $1.5261 \pm 0.0386$ | 11.84% | Conv4 | $0.1991 \pm 0.0001$ | 16.94% |
| Conv5 | $1.4222 \pm 0.0416$ | 11.03% | Conv5 | $0.1958 \pm 0.0002$ | 16.66% |
| MaxPool | $0.0102 \pm 0.0006$ | 0.08% | MaxPool | $0.0005 \pm 0.0001$ | 0.04% |
| FC | $0.3777 \pm 0.0233$ | 2.93% | FC | $0.0077 \pm 0.0013$ | 0.66% |
| FC | $0.0709 \pm 0.0038$ | 0.55% | FC | $0.0017 \pm 0.0001$ | 0.14% |
| FC | $0.0168 \pm 0.0018$ | 0.13% | FC | $0.0007 \pm 0.0002$ | 0.06% |
| Softmax | $0.0028 \pm 0.0015$ | 0.02% | Softmax | $0.0038 \pm 0.0098$ | 0.32% |
| Total | 12.8885 | | Total | 1.1752 | |

Table 1: Evaluation time in seconds per layer on CPU (left) and GPU (right) with batch size of 128. Results are averaged over 8 runs.

## 2 Comparison of distance metrics

As described in Section **??**, we explored explored three different approximation metrics that can be used when finding an approximation $\widetilde{W}$ of $W$. The natural choice for an approximation criterion is to minimize the Frobenius norm between $\widetilde{W}$ and $W$. Our additional proposed metrics are referred to as the (approximate) Mahalanobis distance metric and the data covariance distance metric. Both the augmented distance metrics tend to perform better that the Forbenius norm. However, for different layers, and different approximation parameters, the two augmented metrics often produce different results.

Figure 1 plots the performance of the different metrics on the 50K validation set for the monochromatic approximation applied to the first layer. In this case, data covariance metric produced significantly better results than either of the other metrics.

Table 2 compares the different distance metrics when applied in conjunction with the biclustering with SVD approximation. We applied the approximation to several convolutional layers and tested different hyperparameter settings. We found that the data covariance metric produced the best results. We also explored the effect of the different distance metrics on the biclustering with outer product decomposition approximation technique. Results for different convolutional layers and hyperparameters are shown in table 3. Interestingly, in this case the Mahalanobis distance metric produced the best results.

## 3 Theoretical speedups

We can measure the theoretically achievable speedups for a particular approximation in term of the number of floating point operations required to compute the target output. While it is unlikely that

Figure 1: Comparison of distance metrics for the monochromatic approximation.

any implementation would achieve speedups equal to the theoretically optimal level, the number of necessary floating point operations still provides an informative upper bound on the gains.

Table 4 shows the theoretical speedup of the monochromatic approximation applied to the first convolutional layer. The majority of the operations result from the convolution part of the computation. In comparison, the number of operations required for the color transformation is negligible. Thus, the theoretically achievable speedup decreases only slightly as the number of color components used is increased.

Table 3 and Table 2 outline the theoretically achievable speedups for the biclustering techniques.

## 4   Combined results

We used the monochromatic approximation with 6 colors for the first layer. Table 5 summarizes the results after fine-tuning for 1 pass through the ImageNet12 training data using a variety of second layer approximations.

| Layer | Cluster sizes | Rank of approx. | | Increase in test error | | | Reduction in FLOPS |
|---|---|---|---|---|---|---|---|
| | | $K_1$ | $K_2$ | Frobenius | Data covariance | Mahalanobis | |
| Conv 1 | $G=2$ $H=2$ | 14 | 51 | 13.38% | **7.6%** | 13.35% | 8.6× |
| | | 16 | 51 | 10.61% | **3.16%** | 6.45% | 7.5× |
| | | 19 | 44 | 10.69% | **2.56%** | 10.46% | 7.3× |
| | | 19 | 51 | 6.76% | **1.97%** | 6.44% | 6.3× |
| | | 19 | 64 | 2.39% | **1.72%** | 2.43% | 5× |
| | | 19 | 76 | 1.88% | **1.82%** | 1.9% | 4.2× |
| | | 24 | 64 | 2.19% | **0.69%** | 2.2% | 4× |
| | | 24 | 76 | 1.06% | **0.49%** | 1.05% | 3.3× |
| | | 28 | 64 | 1.82% | **0.44%** | 1.84% | 3.4× |
| | | 28 | 76 | 0.98% | **0.38%** | 1% | 2.9× |
| Conv3 | $G=2$ $H=4$ | 38 | 38 | 13.05% | **4.61%** | 12.85% | 6.5× |
| | | 38 | 51 | 8.21% | **3.02%** | 8.11% | 5.1× |
| | | 38 | 64 | 5.92% | **2.57%** | 5.88% | 4.2× |
| | | 38 | 76 | 3.84% | **2.41%** | 3.92% | 3.6× |
| | | 51 | 38 | 11.65% | **4.04%** | 11.5% | 5.1× |
| | | 51 | 51 | 5.83% | **2.09%** | 5.73% | 4× |
| | | 51 | 64 | 3.69% | **1.25%** | 3.68% | 3.3× |
| | | 51 | 76 | 2.12% | **1.08%** | 2.18% | 2.9× |
| | | 64 | 38 | 11.27% | **3.94%** | 11.04% | 4.2× |
| | | 64 | 51 | 5.19% | **1.81%** | 5.14% | 3.3× |
| | | 64 | 64 | 2.92% | **0.94%** | 2.89% | 2.8× |
| | | 64 | 76 | 1.4% | **0.63%** | 1.42% | 2.4× |
| | | 76 | 38 | 11.08% | **3.85%** | 10.92% | 3.6× |
| | | 76 | 51 | 5% | **1.75%** | 4.92% | 2.9× |
| | | 76 | 76 | 1.21% | **0.46%** | 1.22% | 2.1× |
| | | 76 | 64 | 2.71% | **0.82%** | 2.72% | 2.4× |
| | | 76 | 76 | 1.65% | **0.91%** | 1.65% | 2.1× |
| | | 76 | 64 | 2.4% | **1.22%** | 2.37% | 2.4× |
| | | 76 | 51 | 3.92% | **2.15%** | 3.91% | 2.9× |
| | | 76 | 38 | 8.33% | 4.66% | 8.37% | 3.6× |
| | | 64 | 76 | 2.36% | 1.17% | 2.37% | 2.4× |
| | | 64 | 64 | 2.98% | 1.45% | 2.98% | 2.8× |
| | | 64 | 38 | 8.52% | 4.8% | 8.49% | 4.2× |
| | | 51 | 76 | 4.32% | 1.9% | 4.39% | 2.9× |
| | | 51 | 64 | 5.61% | 2.21% | 5.69% | 3.3× |
| | | 51 | 51 | 5.93% | 2.78% | 5.91% | 4× |
| | | 51 | 38 | 9.19% | 5.18% | 9.25% | 5.1× |
| | | 38 | 76 | 11.62% | 4.29% | 11.65% | 3.6× |
| | | 38 | 64 | 11.37% | 4.45% | 11.49% | 4.2× |
| | | 38 | 51 | 12.28% | 4.52% | 12.39% | 5.1× |
| | | 38 | 38 | 13.87% | 6.14% | 14.06% | 6.5× |

Table 2: Comparison of distance metrics for the biclustering with SVD approximation.

| Layer | Cluster sizes | Rank of approx. | Increase in test error | | | Reduction in FLOPS |
|---|---|---|---|---|---|---|
| | | $K$ | Frobenius | Data covariance | Mahalanobis | |
| Conv 2 | $G = 48$ $H = 2$ | 5 | 10.07% | 11.27% | **6.78%** | 8.3× |
| | | 6 | 6.04% | 7.03% | **4.66%** | 6.9× |
| | | 8 | 3.12% | 3.39% | **2.28%** | 5.2× |
| | | 10 | 1.7% | 1.9% | **1.56%** | 4.1× |
| | | 12 | 0.97% | 1.15% | **0.86%** | 3.4× |
| | | 14 | 0.51% | 0.68% | **0.48%** | 2.9× |
| | | 16 | 0.37% | 0.4% | **0.18%** | 2.5× |
| Conv 3 | $G = 128$ $H = 4$ | 1 | 54.59% | **42.19%** | 56.95% | 16.6× |
| | | 2 | 30.73% | **13.92%** | 27.09% | 8.3× |
| | | 3 | 13.74% | **5.38%** | 13.49% | 5.5× |
| | | 4 | 5.65% | **2.36%** | 5.64% | 4.1× |
| | | 6 | 1.52% | 1.24% | **0.84%** | 2.8× |
| | | 8 | 0.31% | 0.29% | **0.28%** | 2.1× |
| Conv 4 | $G = 256$ $H = 8$ | 1 | 32.87% | **28.27%** | 36.84% | 16.6× |
| | | 2 | 10.2% | **8.77%** | 11.52% | 8.3× |
| | | 3 | 4.14% | **3.41%** | 4.49% | 5.5× |
| | | 4 | 1.65% | **1.59%** | 1.82% | 4.1× |
| | | 5 | 0.8% | 0.92% | **0.86%** | 3.3× |
| | | 6 | 0.42% | **0.58%** | **0.39%** | 2.8× |
| | | 8 | 0.12% | **0.11%** | 0.16% | 2.1× |

Table 3: Comparison of distance metrics for the biclustering with outer product decomposition approximation.

| Number of colors | Increase in test error | | | Theoretical speedup |
|---|---|---|---|---|
| | Frobenius metric | Data covariance metric | Fine-tuned | |
| 4 | 16.11% | 5.94% | 1.71% | 2.97× |
| 6 | 9.9% | 2.39% | 0.43% | 2.95× |
| 8 | 3.51% | 1.41% | 0.22% | 2.94× |
| 12 | 2% | 0.71% | 0% | 2.91× |
| 16 | 1.16% | 0.45% | - | 2.88× |
| 24 | 1.43% | 0.45% | - | 2.82× |

Table 4: Performance when first layer weights are replaced with monochromatic approximation and the corresponding theoretical speedup. Classification error on ImageNet12 validation images tends to increase as the approximation becomes harsher (i.e. fewer colors are used). Theoretical speedups vary only slightly as the number of colors used increases since the color transformation contributes relatively little to the total number of operations.

| Layer 2 | | Increase in error |
|---|---|---|
| Method | Hyperparameters | |
| Biclustering + outer product decomposition | $G = 48; H = 2; K = 8$ | 1% |
| Biclustering + outer product decomposition | $G = 48; H = 2; K = 6$ | 1.5% |
| Biclustering + SVD | $G = 2; H = 2; K_1 = 19; K_2 = 64$ | 1.2% |
| Biclustering + SVD | $G = 2; H = 2; K_1 = 19; K_2 = 51$ | 1.4% |

Table 5: Cascading approximations.