[Reviews · NeurIPS 2014]

Submitted by Assigned_Reviewer_14

The paper addresses the problem of speeding up the evaluation of pre-trained image classification ConvNets. To this end, a number of techniques are proposed, which are based on the tensor representation of the conv. layer weight matrix. Namely, the following techniques are considered (Sect. 3.2-3.5):
1) SVD decomposition of the tensor;
2) outer product decomposition of the tensor;
3) monochromatic approximation of the first conv. layer - projecting RGB colors to a 1-D space, followed by clustering;
4) biclustering tensor approximation - clustering input and output features to split the tensor into a number of sub-tensors, each of which is then separately approximated;
5) fine-tuning of approximate models to (partially) recover the lost accuracy.

Also, the authors consider two approximation criteria (Sect. 3.1):
1) diagonal Mahalanobis metric, defined by the importance of the individual weights, computed by backprop;
2) data covariance metric, which takes into consider the distribution of the layer inputs.

The proposed speed-up methods are applied to the first and second convolutional layers of the state-of-the-art ConvNet architecture [8], and they are shown to speed-up the computation of these layers (but not the whole network) by a factor of 2-3, with a little loss of accuracy. SVD decomposition is also applied to the fully-connected layers of the network to reduce the number of parameters.

== Positive points ==

1. The problem of speeding up the computation of large ConvNets is relevant.

2. The proposed methods for speeding-up ConvNets are novel.

== Negative points ==

The main problem of the paper is that even though a multitude of methods is proposed, their evaluation is superficial, which makes it hard to understand how these methods compare to each other. Also, the effect of methods' parameters is not properly explored. It would be really helpful to have a "take home" message for deep learning practitioners, which provides clear recipes on how to use the proposed methods. For that purpose, the authors might want to add an "accuracy vs speed-up" plot for the whole net (not just two speeded-up layers), with all approximations combined. Also, the description of some of the methods is too concise. A detailed list of issues and questions is given below.

== Detailed comments ==

1. Sect. 3.1: the proposed distance metrics are not compared to each other (and the Frobenius norm baseline) in the experiments. Fig. 4 only contains the results of the data covariance distance metric, the Mahalanobis metric (and the Frobenius norm) results do not seem to be reported. By the way, what is denoted as "Original" in Fig. 4?

2. l.128, (1): which value of h did you use? Why not to compute the derivatives of the correct class score w.r.t. the approximated weights?

3. l.144-153: the description is too concise, it should be extended (especially if this is the method used in the experiments).

4. l.236, l.201 mention that the biclustering approximation is used for higher convolution layers. However, it's only applied to the first two layers. If the approximation was evaluated on higher levels, these results should be reported; otherwise, these statements should be corrected.

5. l.236-242: is it true that each sub-tensor corresponds to a subset of input-output channels, and the subsets do not overlap, i.e. each channel is "covered" by a single sub-tensor? The authors should elaborate on how the result is computed by including a formal equation, instead of just referring to Fig. 1(b).

6. l.241-242 ("each sub-tensor contains similar elements"): do you mean similar in the sense of the distances (2) or (3)? Are there any theoretical reasons why each cluster lives on a low-rank manifold, and is thus easier to approximate than the original tensor?

7. Sect. 4.1.2: two approximations of Sect. 3.2.2 are not compared to each other without biclustering. It would be helpful to see the benefits of biclustering.

8. l.362-365 ("we explored various configurations"): only the results corresponding to 2 combinations of G and H are given in the paper and the supp. material, which is not sufficient. In Fig. 4, why are different values of G, H, K, K1, and K2 used for the CPU and GPU implementations? How are they chosen?

9. l.376 ("performance is restored"): there is no guarantee that the performance can be fully restored by fine-tuning. It would be helpful to see the results with and without fine-tuning.

10. Sect. 4.3: how does SVD on the fully-connected layers affect the evaluation speed?

11. l.269 ("4 convolutional layers"): isn't it supposed to be 5, if the configuration of [8] is used?

12. l.285 ("20K validation images"): why not to report the performance on all 50K validation images, as typically used in the literature?

13. Speed-up of the whole net: it would be helpful to see the speed-up of the whole net, rather than just the first two layers. According to Table 1 of the Supp. material, it seems that x2 speed-up of the first two conv. layers will result in only 22% reduction of the evaluation time of the whole net on the GPU?

14. Relation to [6]: it is understood that this work was developed in parallel to [6] (recently published at BMVC-2014). While the comparison with [6] is not strictly required, it would be interesting to see it in the final version, if this paper gets accepted. Interestingly, [6] report 2-4 times speed-up for the *whole* network, but obviously the ConvNet configurations considered in this submission and [6] are different, so a direct comparison is not possible at the moment, and the authors are encouraged to add one.

== Typos ==

l. 214: "consider consider"
l. 211, l.315 "Figure 4.1.1." -> "Figure 2"
Summary: The paper discusses a relevant problem (speeding-up ConvNets at test time) and proposes a number of rather novel methods to address it. On the negative side, the methods are explored and compared somewhat superficially, so the paper has a lot of potential for further improvement.

Submitted by Assigned_Reviewer_22

This paper introduces various ways of reducing the number of parameters of CNNs for image classification using tensor decomposition methods. This test time speed up speed ups are a factor 2 to 3 with little reduction in performance.

I have one technical question for the response period. It seems to me that each approximation method corresponds to some fixed architecture with linear layers. It seems to me that back-propagation (gradient descent) can be done in networks with linear layers. So the question is, what happens if we simply train the reduced network with the linear layers from the start. For example, the monochrome approximation corresponds to a first layer that constructs a set of monochrome channels (without nonlinearities). There is one monochrome channel for each first-layer feature. Each feature is then a scalar CNN over its channel. If we fix this architecture and train it then we should get a speed-up in training time as well as test time. No? Have you tried it?
Summary: This is an interesting paper with interesting experimental results. I believe it should accepted largely because of the intense interest in CNNs at the current time.

Submitted by Assigned_Reviewer_43

# Summary
The paper describes strategies for factoring weight matrices in CNNs to increase their (test time) speed and reduce memory requirements.

# Quality
The proposed methodology seems sound.

# Clarity
The paper is clear in the description of the method and the results.

# Originality
The authors note that their techniques are closely related to a few other pieces of recent research.

# Significance
Speeding up test-time performance is an important part of large-scale deep learning
application in industry. Techniques like this aren't often described in academic literature.
Summary: The paper is sound, and clear. Speeding up test-time performance is an important part of large-scale deep learning
application in industry; and such techniques are not published.

Submitted by Assigned_Reviewer_44

Summary:

The paper proposes a number of techniques to compress, or approximate, the (trained) parameters of lower convolutional layers of a deep convolutional neural network. The authors show that the proposed approximations lead to computational speed up and memory reduction during test time. In doing the approximations, the authors propose to use non-Euclidean metric for the distance between the original matrix (or weight tensor) and the approximation.

Originality, Quality and Clarity:

Often people tend to focus more on speeding up training rather than inference when it comes to deep neural networks, although recently there have emerged some works (most of them cited in this paper) focusing more on inference time speed. In this sense, I find this paper to be fairly original. Also, the paper validates the proposed methods on both CPU and GPU, which adds to its contribution.

The paper is well written and in most parts clear to follow. However, the explanation of each proposed method is too densely packed and not easy to ‘decipher’ what precisely each approximation does, which I suspect is due to the limitation on the length. I’d appreciate it, if the authors ‘unfold’ the explanations of those methods in the supplementary material.

Significance:

The direction toward which this paper is heading is important. More mind-blowing speed up and memory reduction would’ve made this paper more significant, but I think 2x speed up and 3x reduction is still pretty good.

Other Comment:

One minor, but a bit contradictory point I find in the paper is on finetuning. At the end of Sec. 3.5, the authors mention that ‘the upper layers can be fine-tuned until the original performance is restored’. But, as you show in the experiments and as expected with a lower number of parameters, you’ll essentially never get to the original performance, won’t you? I guess you simply do early stopping with validation set, don’t you?

Another minor suggestion that might have been interesting is to train a model with the reduced parameter set from scratch. Of course, because of the clustering involved, I highly believe that you’ll get nowhere with this, but it may be interesting to see and be a good control experiment (if you can get somewhere close in accuracy to what you get, why wouldn’t you start training from the reduced model?)
One last remark: where did you start finetuning from? From ‘Original’ or ‘||W||_data distance metric’?

Few typos:
- line 97: ‘Out work’ => ‘Our work’
- Eq. (1): I think you’re missing one $i$, no?
- line 211: ‘Figure 4.1.1’ => ‘Figure 1’
- line 422: ‘translate’ => ‘translates’
Summary: This paper introduces a set of approximation methods to speed up the inference of a deep convolutional neural network, which resulted in roughly 2x speed up and 3x memory reduction. The research direction in which the authors are heading is important and interesting.
Author Feedback
Author rebuttal: We thank the reviewers for their particularly constructive and relevant comments.

(R1) Our main goal is to obtain significant speedups of large, deep CNNs at test time, that practitioners can directly deploy on standard CPU and GPU architectures. For that purpose, we worked on two research fronts: (i) developing efficient linear (and piecewise linear) compression strategies of convolutional tensors, and (ii) optimizing them on both CPU and GPU to measure practical speedups. We agree that our evaluation across compression parameter space and all layers of the network is not exhaustive. We were limited by space and thus tried to emphasize key results. However, we do have data comparing different approx. criteria and wider parameter ranges, that we commit to report in the final version.
Another point is what happens beyond the second layer. We concentrated our efforts on the first two layers, since they take up most of the complexity at test time. Preliminary results on higher layers suggest that they admit similar compression factors, but we agree with R1 that a more thorough analysis is required. We will provide more extensive whole network speedup results in addition to individual layer evaluations.

(R1) #1: The efficacy of each metric differed based on the layer and approx. type. Due to space constraints we plotted only the metric which worked best for each approx. type/layer. We will add a fuller analysis to the final suppl. material. “Original” refers to the frobenius norm without any finetuning. We will clarify this in the final version.

(R1) #2: We used h=3. The goal of our proposed metric is to emphasize directions that might cause the network to tip to the wrong label, but indeed, the derivatives of the correct label might also be used to estimate the covariance, since they also influence the predictive performance of the network. However, since the network is optimized by following these derivatives, at convergence the weights are at a local minima, where those gradients are zero, so we expect those directions to have small influence in the covariance estimation.

(R1) #4: We focused on the first two layers since they take up the majority of inference time. We will add a further analysis of higher layers to the supp. material. We agree that the text should be corrected to reflect our current experiments.

(R1) #5: Yes, the subtensors do not overlap. As described in section 3.4, we independently cluster the rows of W_C and columns of W_F. We use k-means in all our experiments. We also tried subspace clustering, but found that it wasn’t significantly better and so we opted for the simpler, less computationally intensive method.

(R1) #6: In the cases where the augmented distance metrics were used, yes, we mean (2) and (3). Otherwise, just Euclidean. Each bi-cluster is a convolutional tensor of much smaller dimensionality than the original, which is constructed to contain similar input/output features. As a result, each subtensor is compressible with linear models of low rank (less than 10 in experiments). For speedup purposes, it is important to break up large dimensions (256 output features) into small groups.

(R1) #7: Without the biclustering the approximations are mostly infeasible.

(R1) #8: We explored a large variety of G and H but plotted only a handful of reasonable ones due to space constraints and to keep plots easily readable. We will add further explorations to the supp. material. We implemented the outer product decomposition approx. on GPU and SVD decomposition approx. on CPU because we found the different implementations to be more amenable to different computing paradigms. However, we see no theoretical reason why both types couldn’t be implemented on both the CPU and GPU.

(R1) #9: See figure 3 and 4. The blue points refer to the results without finetuning. Also the supp. table 2 shows more detailed results of the effect of finetuning for the first layer.

(R1) #10”: As shown in supp. table 1, the FC layers take negligible amount of time to begin with so we did not test this.

(R1) #11: This is a typo, thanks for pointing it out.

(R1) #12: Also a typo, we in fact did use the entire 50K set.

(R1) #14: We would like to emphasize that our results are on a challenging real world problem. The ImageNet architecture is significantly larger than that of [6] making it a much more challenging model to obtain speedups on. That said, we agree that full network comparison on same architecture (and same task) should be performed. We’ll look into this.

(R1,R4) Description of approx. techniques is too concise: We agree with this point and will gladly add lengthy description to the supp. material.

(R2, R4) What happens if we train the decomposed network from the start?: We hypothesize that restricting the network from the start (i.e. enforcing each layer retain it factorized form throughout training) is interesting from a speedup perspective and plan on exploring this in future work. However, redundancy in highly nonconvex optimization problems might help avoiding local minima at early stages.

(R4) “as you show … never get to the original performance” Our main aim was to get below 1% drop in performance, which we consider negligible. Further finetuning could likely drop performance lower, however we stopped after about one epoch through the training set since this was sufficient for our purposes.

(R4) “where did you start finetuning from?’”: The results reported start from ||W||_distance. We also tried starting from original on a couple cases. This took longer to fine-tune and so we abandoned it since the ||W||_distance starting point worked.